# The Media in the Construction of Reality in the Context of Colombian Social Nonconformity

Andrés Barrios-Rubio [1,2,*] and Gloria Consuelo Fajardo Valencia [3]

1   Faculty of Communication and Arts, Antonio de Nebrija University, Hoyo de Manzanares, 28240 Madrid, Spain
2   Department of Communication, Pontifical Javeriana University, Bogota 56710, Colombia
3   Faculty of Social Sciences, Jorge Tadeo Lozano University, Bogota 110311, Colombia
*   Correspondence: andresbarriosrubio.abr@gmail.com or barriosr_andres@javeriana.edu.co

**Abstract:** Introduction: Convergence of linguistics and semiotics materializes in the text not only the conceptual content that is expressed through codes, but the message also underlies the realism of the communicative intentions of the issuing agent in a specific context and the value of the interactions of the actors of the communicative act. Methodology: The vision of reality that is established in the collective imaginary must be analyzed from the interpretation of society and culture, the laws of operation and their constituent parts. For this, this research approached two newspapers of national circulation, five general radio channels of national coverage, and two television newscasts of private channels, and through the use of quantitative instruments, the posts, tweets or videos were reviewed in order to analyze the constituent elements of the discourse—text, images, hashtags, or keywords—which are appreciated from the syntactic and semantic perspectives (structural) and pragmatics (functional). Results: The communication process in its social context denotes the intervention of nonlinguistic elements of sociocultural order that demarcate the generation and interpretation of the meanings and senses of linguistic expressions. Discussion: The linguistic structure offers conditions for communication, but any generation and transmission of meanings is a product of the intention of the subjects who use it for specific purposes and in specific communicative situations within a social context. Conclusions: Intensive use of digital technology and social networks naturalized a relationship of proximity and familiarity in the communication process, satisfaction of needs in the multiplicity of information that is created and distributed in the network bundled to mobile devices and the political and social ecosystem of the nation.

**Keywords:** discourse; social reality; media; messages; communicative act

## 1. Introduction

An environment of nonconformity marks the deepest political chaos in the democratic history of Colombians. The nation faces a complex agenda of economic and social problems [1] which made it clear that a breach of agreements is what has generated more violence, misery, unemployment, and abandonment towards a people that demands opportunities (Figure 1). The dissatisfaction is great, the public debate occurs in divergent scenarios: social conversation on the street, social networks, media, the family; scenarios marked by opportunistic strategies, pointing out, disqualifications, and personal assaults that increase on digital platforms [2] and lead to vandalism, anarchy, roadblocks, extortion, chartering, among other factors. The Havana Peace Agreement, the 2016 Plebiscite for Peace, the implementation of what was agreed with the FARC, the SARS-CoV-2 pandemic, and the economic crisis have been a source of discord. Speeches, statements, and messages show that the dynamics of political and social action are as contradictory as the positions of the actors of the Colombian ruling and media class [3].

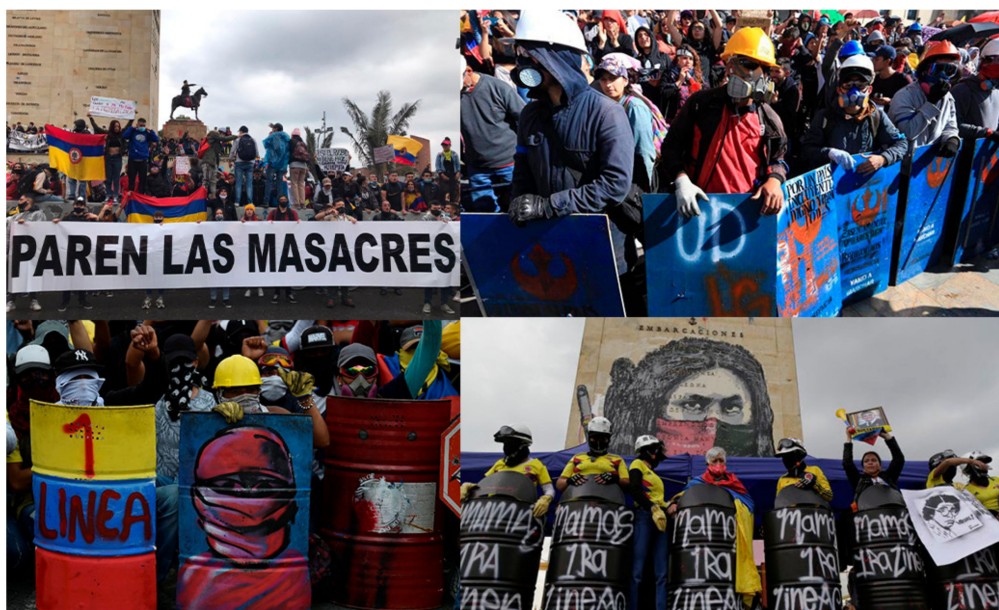

**Figure 1.** Images replicated by the village media in the street. Source: Own elaboration with images of social networks. * The image shows the young constituting urban cells that are known as "front lines" and the clamor of the mothers asking the forces of order to stop the massacres against their children in the midst of social protests.

A polarization and crisis environment that accompanies the Colombian social environment invites us to reflect on ways of interpreting the world [4] and understanding the role of semiotic and discursive elements in the construction of meaning [5]. From communication, questions are woven that build a look and interdisciplinary alliances which determine the consolidation of thought and knowledge against a theme of the content agenda [6]; meaning from ICTs, meanings today mediated by the image that the media have established imperiously in everyday life [7]. The periodic referents, present in the life of the subjects, are reassessed and reconceptualized from values or categories that rise from languages, textual and contextual, that raise levels of abstraction that are evidenced in the intellectual landscape of the citizen collective.

Semiotics is the articulating axis of the message, language, and its contents [8], a phenomenon that from the sciences of communication leads to study the successive, circular, and dynamic chain of elements that show a variable world, with times and spaces immediate by the characteristic of instantaneity that accelerates with the dynamics of the digital ecosystem [9]. It is leading a circumstantial biosphere in which communication without significance is palpable [10], the interpretation of reality by the citizen is biased by the code that underlies the text and that must be understood from the interpretation given when the assessments that each author gives to the concepts are found [11] and the intertextual relations are elaborated [12], and the cultural context to which it is accessed [13], among many other variables.

The symbolic universe built by citizens raises new ways of knowing, under language lie a series of events, objects, and cultural ways of interpreting a reality embodied in everyday interaction [14]. The meaning of the messages is re-evaluated, surrounding the image and its social preponderance, knowledge and information are internalized through the senses. Correlation of the image with the word is established in the cognition of individuals from their disposition in universes where the epistemological and communicative boundaries between signification, representation, and interpretation become diffuse [5]. The relationship of subjects with objects has generated cultural spaces full of meaning [15], configuration of the universe that surrounds the person, and movements that present regularities and irregularities related to the concordances or agreements reached by the social group.

Language explains the pertinent, the impertinent, the spatial, the temporal, the logical, and the illogical, constructing a conglomerate that supposes intelligence structures that define the person as a subject. ICTs with their ways of perception, meaning, and interpretation have set their sights on the way images are represented on the stage of new technologies [16]. Perception requires before such a subjugation several times and spaces that relate one after another and that not only reflect faster and closer interactions with the other but that are associated with the symbolic [17]. The discussion on communication and information begins to measure the basis of human activities and world conceptions, intelligible messages need to be structured in a form that appears as redundant and has multiple elements as repertoires of meaning found in the senses, originality, and uses as input for actions.

Attribution of significance to a signal by the receiver depends on the indications [10], those perceptible facts that refer to anything that is not, and the signals, conventional indications that are produced expressly to manifest to the listener, hearing or seeing an intent from the sender. Study of the messages discusses the interaction, mediation, and interpretation in the communicational products, a scenario in which significance crosses them, forming a complex node that is good to define. In the research that is now described, the semiotic field was considered as a problem nucleus or work perspective. The accesses of users to ICTs, new ways of informing themselves, and the types of rituals that accompany them were recognized. The objective is to show how information is discriminated against and how its social interaction has changed with the use of symbols that are handled on the internet.

The research process was articulated according to three purposes, framed within semiotics, namely:

Q1: New ways of knowing and evidencing that knowledge of a fact.

Q2: Cultural objects and goods, their relationship with the world, and the meaning of those things.

Q3: Relate significance to social interaction, things, and their meaning.

Speech as a linguistic communicative unit [18], oral or written, par excellence is an indivisible global module loaded with meaning from certain components: who, how, why, and when language is used in a communicative context, with an intensity, some beliefs, and an intensity of social interaction from a knowledge that the issuer has. The process of enunciation involves the actors of communication in a social, cultural, ideological, and physical context [19]; the interaction between the participants of the communicative act not only reflects advents of speech, marks the pronouns, the verb tenses, the deistics, marks that allow an interpretation from a relation of the text with the context. Discourse theory [20] allows articulating textual forms or text schemes that determine the production of the speech. In the light of the enunciation, it becomes relevant to address and analyze the relationship of the processes involved, in the construction of meaning of the discourse as they are: action, context, power, and ideology, arguing the power of significance of this linguistic unit [21].

## 2. Methodology

Based on the General Media Study EGM [22], we can see that the Colombian communication industry in its conventional and digital offer has an important presence in the consumer agenda of citizens: traditional press—74.7%, digital 52.9%, radio—traditional 83.4%, digital 59.6%, traditional television—91.6%, digital 64.1%. The informative diet of the public is related to the journalistic brand of the medium and becomes relevant in social profiles on Facebook, Twitter, Instagram, and YouTube (Table 1), the epicenter of convergence and interaction where the interests of the sender and the receiver are intermingled. To delineate the construction of reality in the midst of social nonconformity, it was determined to take as a basis of study two newspapers of national circulation, El Tiempo and El Espectador, five generalist radio channels that broadcast for the whole country from Bogotá—Caracol Radio, W Radio, Blu Radio, RCN Radio, and La FM—and

two television newscasts of the private channels, Noticias Caracol and Noticias RCN, in the period between 1 April and 31 May 2021.

**Table 1.** Conventional media, its users, and publications on digital platforms.

| | **Facebook** | | | |
| --- | --- | --- | --- | --- |
| **Media** | **Followers** | | **Posts** | |
| | April | May | April | May |
| Press | 10,255,482 | 10,285,524 | 7006 | 7100 |
| Radio | 6,404,900 | 6,418,250 | 14,825 | 14,600 |
| Television | 11,114,864 | 11,178,518 | 3746 | 3490 |
| | **Twitter** | | | |
| **Media** | **Followers** | | **Posts** | |
| | April | May | April | May |
| Press | 12,947,360 | 13,212,412 | 14,576 | 15,038 |
| Radio | 16,546,950 | 16,748,515 | 26,905 | 25,065 |
| Television | 7,080,202 | 7,082,666 | 3934 | 4208 |
| | **Instagram** | | | |
| **Media** | **Followers** | | **Posts** | |
| | April | May | April | May |
| Press | 3,838,436 | 3,957,668 | 816 | 864 |
| Radio | 2,894,290 | 2,956,025 | 1335 | 1430 |
| Television | 5,457,262 | 5,500,106 | 328 | 364 |
| | **YouTube** | | | |
| **Media** | **Followers** | | **Posts** | |
| | April | May | April | May |
| Press | 1,630,000 | 1,730,000 | 890 | 958 |
| Radio | 815,000 | 853,800 | 785 | 785 |
| Television | 11,310,000 | 11,550,000 | 470 | 330 |

Source: Own elaboration. * For the development of the study, journalistic houses were joined by industry.

Analysis material triangulates qualitative methodological factors—discourse, images—to address the objective and research purposes [23]. Study of the messages involved reviewing the communicative proposal of the medium and the interaction of users with the post, tweet, or video; public communication of events that position themselves in the collective imaginary [24]. Content analysis details the elements that make up the message [25], speech that contextualizes the event [26], still and moving images that recreate the situation, hashtags or keywords that highlight and position the subject; components that predominate in the corpus collected in the media accounts under study (Table 1).

Critical analysis of the situation [27] is structured from a cross category (Table 2) that exhibits the concurrence of semantic codes in the communicative proposal of the issuing agent. The investigative work is profiled and legitimized with the sketch and realization of the investigation, convictions and reasoning, and assumptions [28]. Discussion of a social reality that materializes with the messages of the media, the positions of public actors, and the reaction of users to the approaches exposed in the profile of the journalistic industry.

**Table 2.** Qualitative analysis variables.

| Category | What He Is After | Criterion of Analysis |
|---|---|---|
| Materialization of the fact | Discursive structure with which the sender reaches the receiver to give reason and meaning to the events of the information agenda. | Context of communication: order, form, sense, and style [26] that give structure to the social situation and allow to understand the discursive act [29]. |
| References that structure the narrative | Reality versus the fiction of events, truth versus the preconceptions of citizens. | Discursive text and its companions to construct the notion of reality in the receiver, units that allow the production of meaning [10]. |
| Receiver interaction | Materialization of citizen thinking and its reactions to the thematic proposal circulating in the social network. | Interpretation of the event under a context determined by the social conjuncture [30]. |

Source: Own elaboration.

The levels of significance exhibit the tactic deployed by the media industry to become part of the citizen conversation in the perception of a reality altered by social nonconformity [3] and characterize the behavior of agents of the communicative process [31]. The multiplicity of broadcast, transmission, and reception media present in the market, added to an audience immersed in screen devices, force to think about the narrative structure that is used to impact the construction of the notion of reality in the collective imaginary.

## 3. Results, and Discussion

### 3.1. Interpretation of Reality from the News Agenda

The corpus of study seen from structural semantics [32] allows establishing the structures of significance and the way in which human experience is organized in conceptual units. The notion of meaning extends to the entire universe of cultural experience and exceeds the level of the real state of events, which is the field of study of a theory of truth or a theory of reference (Figure 2). Although expressions can designate objects or states of the world, in principle, expressions transmit cultural contents, expressions that are understood by their interconnections [32]. The devices that were observed, in Figure 2, segmented the human experience from two principles: community and opposition. Citizens interpret natural and social reality by establishing differences between entities that are related to each other: nuclear semas, invariable nuclei of significance that constitute the definition of terms, and contextual semas, significance components that determine possible contexts.

Figure 2 shows the construction of reality by the media from textual, visual, and audiovisual elements, confrontation of the arguments of the governmental institutionalist with the union positions of the unions and the popular classes. The real situation marks a complex environment marked by a third peak of the pandemic, the need and hunger of the population, and the stubborn pettiness of leaders who encourage crowds and take the citizen to the streets. The polarization mood that surrounds the country shows clear conceptual and ideological differences between the actors of the extreme left against the policies of the right that the government exercises. The notion of reality is marked by the media agenda that is constructed with short and direct headlines of the facts, the image of primary sources and discursive construction from the questions that are all asked within the population collective.

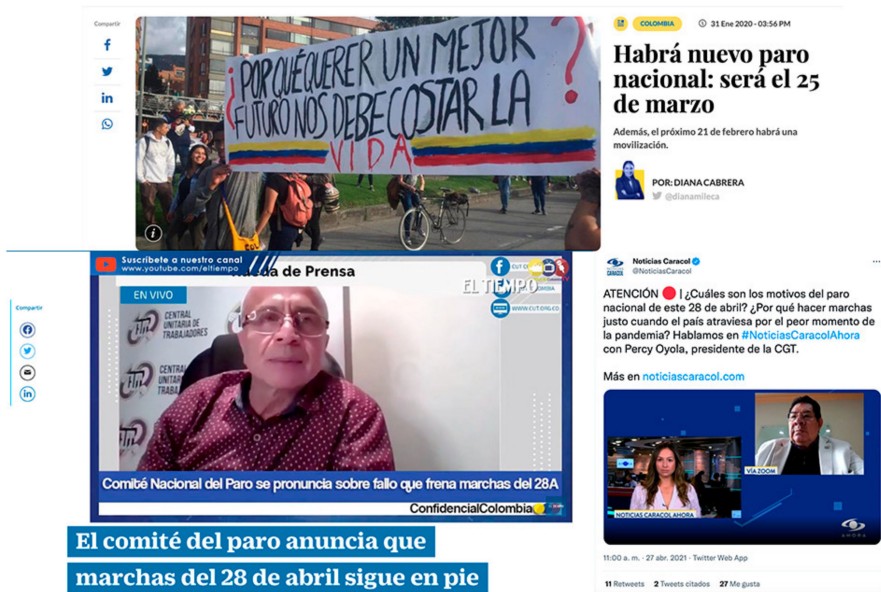

**Figure 2.** News agenda addressed by the media. Source: Own elaboration with images of the web and social networks of the media. * The image shows the promoters of the national strike promoting the marches, despite the legal prohibition that weighed on them, excitement that leads to the street, and the why to want a better future must cost us our lives?

The meaning of words and sentences is composed of semas that come from different semantic fields. The semas are organized from axes that are the ones that generate the networks of relations and oppositions. Semantic fields and semas are the schemes and figures that operate as conditions to apprehend reality and account for the perception of the natural and social universe. Figure 3 denotes a series of immanent categories, prior to the linguistic manifestations expressed in the communication: #EsElMomentoDeParar, #ParoNaciona21A, #ParoNacional21M, #ParoNacionalIndefinite, #NoALaReformaTributaria, #Cacerolazo28A, #SOSPortalAmericas, #SOSColombiaDHH, #PrimeraLineacol, #PortalDeLaResistance, among others, materialization of the suffering of the popular classes in a struggle against the police forces and ESMAD.

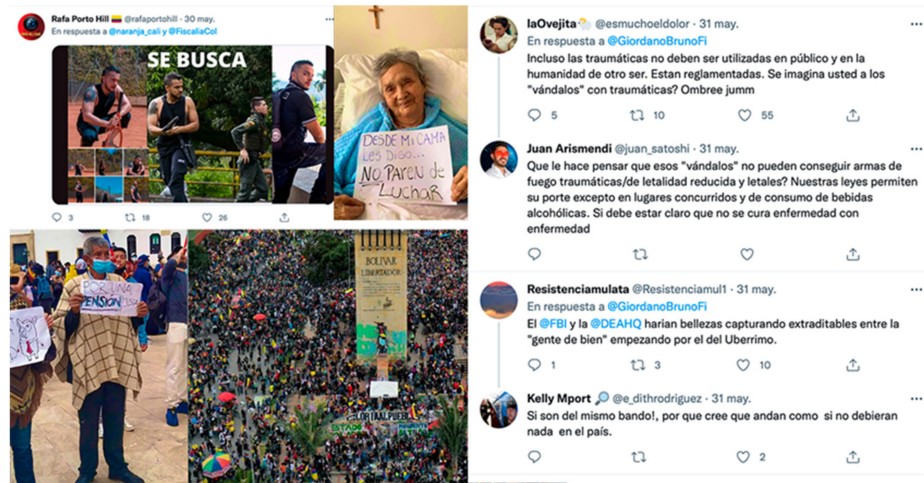

**Figure 3.** Citizens' demonstrations in media messages. Source: Own creation with images from social media networks. * The image shows older adults supporting younger people to keep fighting, a reality that contrasts with armed youth supporting law enforcement to regain normalcy. Ideological confrontation that moves from texts in social settings to armed confrontation in the street.

Semas are organized in lexemes, words, or terms, which is called generative travel. On the one hand, the upper strata of the population invite us not to lower our arms and continue to fight, on the other, there are those who tire of the blockade and seek to take justice into their own hands, a class struggle that opens confrontation and fuels accusations between them; hate speech takes over social settings. The meaning is established in the expressions, from the choices made in the structure of meaning, that is, the skeleton of the content plane meets the framework of the expression plane in which the lexemes and sememas acquire significant materialization according to the following scheme (Figure 4):

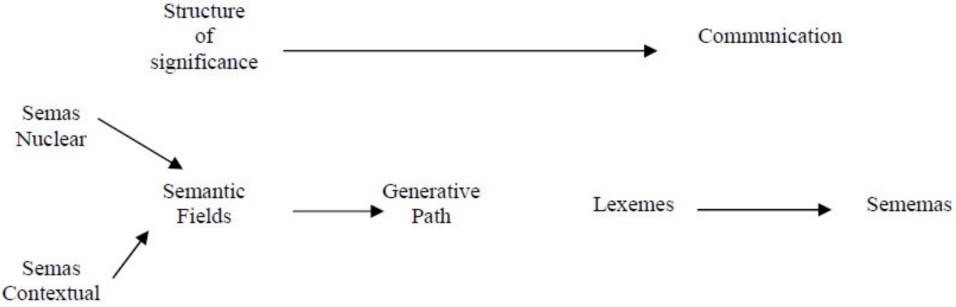

**Figure 4.** Outline of significance. Source: Own elaboration with the structure of Greimas 1971 [32].

What makes the structure of the language is not the units, but the rules and relations; the linguistic activity is supported by a set of underlying rules that enable it. The discussion on the polarization of the population from the publications of the media and its users on social platforms will mean replacing the domain of the observable: excesses in the legitimate right to protest, both of the citizens and of the public force, the experimentation by inference—the bad ones are those of the institutionalist, they abuse the authority and from the excess of the force they are killing the protestant population—and the explanation. In the conflict, there are two sides, in the early hours of the day in peace the people express themselves, in the course of the day, armed actors infiltrate, and at night, the police react to resume order (Figure 5). From half-truths, reality is constructed, and radical thinking for and against unemployment increases.

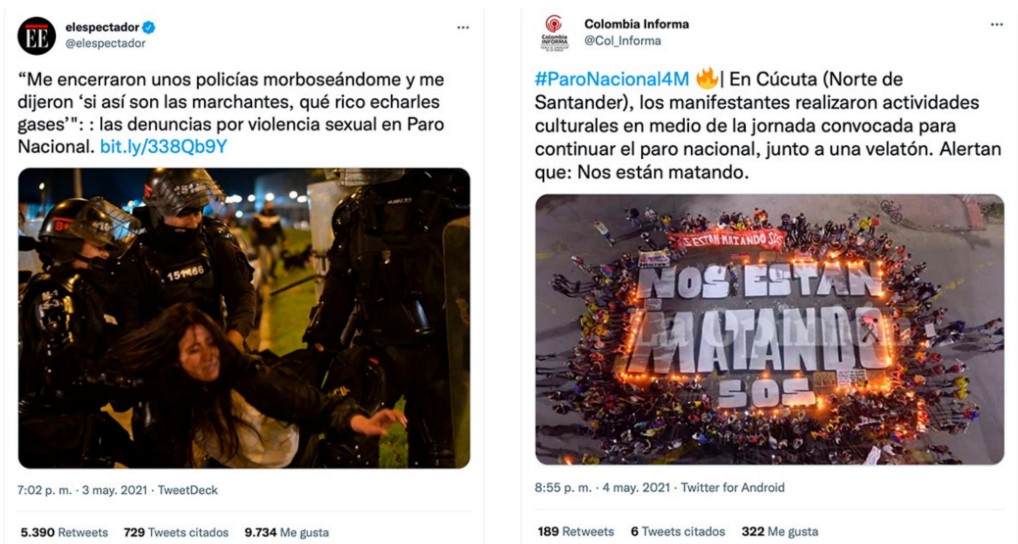

**Figure 5.** The citizens are victimized in the excesses of the protest. Source: Own creation with images from social media networks. * The image shows reactions of young people who incite violence, and when they are detained by the police, they become victims who seek the support of the other demonstrators. In the collective imaginary, the forces of order are exceeding their functions and kill civilians to maintain social calm.

Narratology calls for a review of the structuralist bases of immanence and expression, observing the stories from anthropological, literary, and other fields related to the textual production logics of mass communication: journalism, cinema, graphic arts, among others. The narrative text [33] demonstrates, in the figures exposed in this section, that the messages have a similar structure consisting of a series of immanent logics that materialize in the various narratives. These theoretical contributions bet on giving explanations to the super structural strategies to make clear the underlying processes and logic.

*3.2. Key Categories of the Narrative Text*

The messages circulating in the digital ecosystem allow us to observe levels of the narrative text. In relation to a possible world, it corresponds to the natural, social, and collective reality, in turn, it is plausible when it is based on that reality, but it modifies its schematics as a construction of the text itself (Figure 6). The possible world operates as the basis of the contract of meaning that the text proposes, and its readers-users recognize and assume within the competences of the interpretation. Within the framework of a polarized country, such as Colombia, the formation of armed urban cells that are sponsored by political groups is validated (Figure 6) legitimation of citizen violence and confrontation against the institutionality represented by the government and law enforcement officials. The thematization from its axiological proposal supports the meaning that is wanted to build is based on the choice of the actantial spheres, subjects, and objects of desire, and of the actants, places of the story occupied by characters or aspect of characteristics and environments.

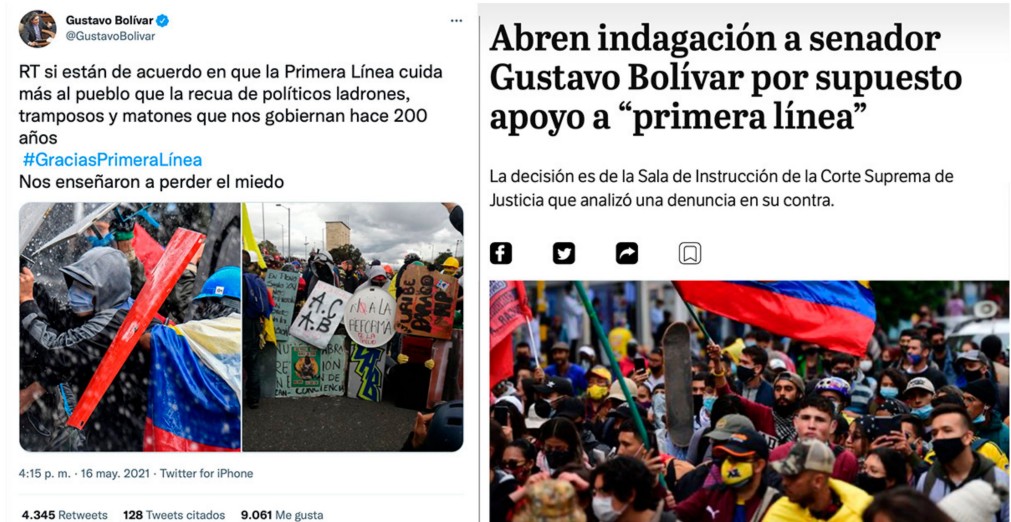

**Figure 6.** Materialization of the reality lived in the street. Source: Own creation with images from social media networks. * The image shows the consolidation of violence in Colombian streets, the formation of urban cells known as "front lines" to replace the police, a social phenomenon that is seconded by left-wing political forces and leads to the opening of judicial proceedings that investigate their actions.

There is no correspondence between actants and characters, since the latter are part of the level of the story, and an actantial sphere may be occupied by several characters, or a same character may occupy several actantial spheres, in the passage from one level to another [34]. Figure 6 highlights the implications of a member of the Senate of the Republic, a member of the opposition, inciting violence and promoting the formation of urban cells. Context of the situation translates functions that determine the status of the actions as nuclei of pure tasks and catalysis, expansions of the foci. The situation reflected here denotes the precept of being in signs—a Senator of the Republic, who promotes the laws and represents the democratic order, sponsors events that lead to violence in the streets, implicit data—proceedings outside the law by an opposition politician are

denounced before the Supreme Court of Justice and an investigation process is opened, and informants—ideological confrontation and interpretation of the circumstances among the followers of left and right currents, explicit details.

Figure 7 exalts the temporal manipulation in which the story organizes in a semiotic time narrative, the logical and chronological time in which the event takes place in the possible world, through anachronisms, relations between order, the duration and frequency of the events being narrated, and the corresponding events in the possible world. The analysis of the corpus of study reveals that the media concentrate the materialization of events in violent acts, the confrontation between demonstrators and law enforcement officials, but the peaceful moments of the march or cultural acts that accompany the expression of the youth at the beginning of the day are invisible; it is important to show that the true sense of the protest is blurred with the infiltration of actors outside the law. A textual strategy is observed that assumes the treatment of history and defines the degree of knowledge and participation in events. The point of view establishes a difference between the instance that narrates and who receives, because the possible world, on which the story is built, is only accessible through perception, whether physical senses or feelings and emotions. This is a point of disruption between the media and audiences, because, in the coverage of the Colombian social conflict, the citizens involved in the marches perceive that in covering the events, the political and economic interests of the agents of the media industry prevailed.

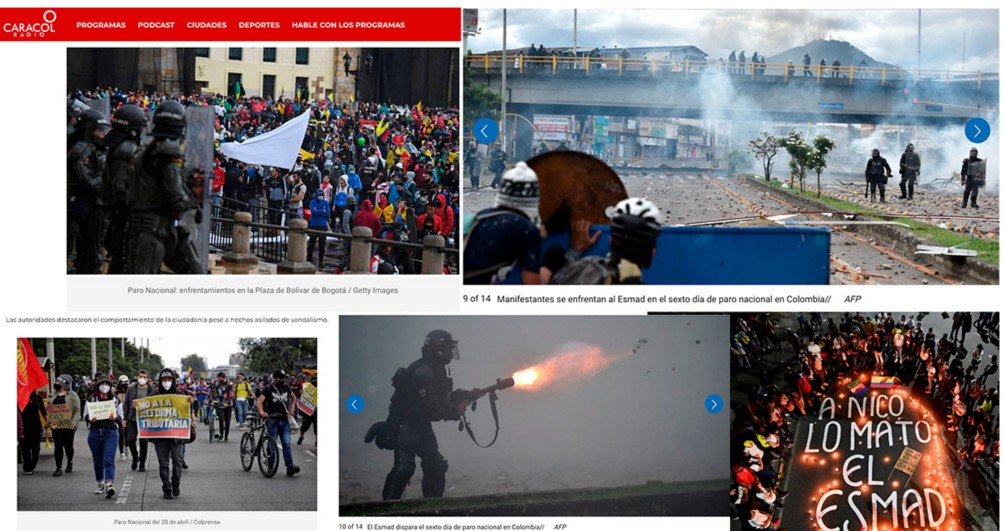

**Figure 7.** Reality and appearances during conflict. Source: Own creation with images from social media networks. * The image shows different moments of the protest, its development throughout the day, the calm of the morning, the heated spirits throughout the day, the violence unleashed at the end of the afternoon, and the return to calm at night.

The text—written, visual, sound, or audiovisual—is the materialization of the choices of the producing instance through which narrative techniques and resources are imposed. The order in which the levels are presented is purely methodological, because in the production and understanding of the communicative product these schemes are updated simultaneously, since the categories and levels are structural, supportive, and dependent on each other. The sign, its meaning and significance, denote the similarity, contiguity, and causality from which the subject can associate ideas that favor judgments from a particular emotion. The study material collected and analyzed allows evidence of a mixture of photographs and suggestive texts in the social scenarios that are reinforced with live sequences from the scene of the events, journalistic input that leads to the expansion of facts through a narrative sequence in which web media, app media, and social media

converge [35]. The sign, seen from Pierce [30], is an emotion, and as preached, accompanies the interpretation that is made of the world.

### 3.3. The Notion of the Interpretant

Reading the social and cultural processes entails questioning whether that which is socially constructed accounts for what is agreed in terms of emotion; triad and diasdic relationships account for the way of knowing what is known. The notions of reality of Colombians are permeated by the ideological interests and political positions of the leaders of the left and right, public statements in the media, and the reactions of each other on social platforms. From the notions of representation, object and interpretant are established the way it is meant in the relations that arise from the interpretants that relate to the same object of the communication. A thought or world model is a mental action that guides the interpretations of the human being and the power exerted by the citizen on them; what is seen in the messages is derived from what is known, the objective reality of the external world (Figure 8). Distrust of the citizen collective with law enforcement agents creates climates of tension and discussion that signify the use of weapons, force, and intimidation in interaction at a time of crisis.

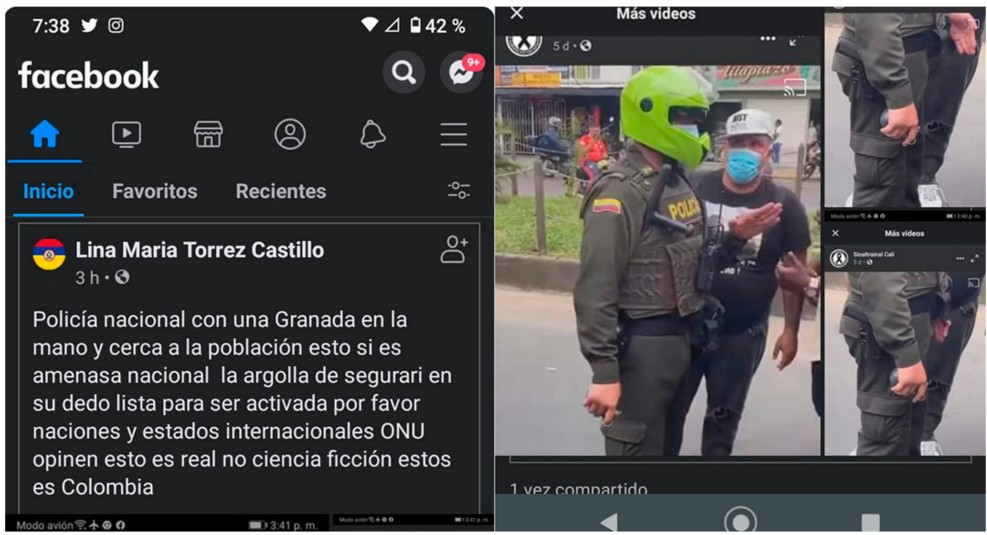

**Figure 8.** The truth in the world of the citizen. Source: Own creation with images from social media networks. * The image shows the delusion of persecution of young people who try to prove that the police want to kill them—an agent talks to citizens while carrying a grenade in his hand.

Inferences, inducements, deductions, hypotheses, or measurements that are derived from the message are linked to the reasoning and knowledge of the person; the context of social reality seeks to reaffirm that the Police are murdering citizens without reason (Figure 8). The thought is considered something internal, something intimate, which allows to handle a series of elements to boast a communication both intimate and social, even in cases of solitude, a communicational structure is presupposed by just belonging, being a part of, or having a membership in a society. Analysis of the interpretation and intentionality of the essential attributes of the message reveals that thought is the meaning, that is, the production and interpretation of signs, potential to be expressed and its susceptibility to be understood. Clashes between the security forces and demonstrators, tear gas and potato bombs that cause injuries and leave members of each social circle flogged are the flow of one thought preceding another in a successive manner, resulting in a flow of thoughts following each other bound by previous cognitions, experience that has a time and is a continuous process called an event, and from there, they are configured and linked together.

Each thought consists of a metaphor, an assumption that is determined from what can be perceived through the senses. It is limited by the frames of reference that are elaborated in the experience. Every decision made from an inductive, deductive, or hypothetical thought can at some point become a conviction, but subsequently, depending on internal and external obstacles, can be converted into an opinion. In Colombia, right-wing opponents are rebel vandals fighting for a cause without reason, while those on the left see in their adversaries arrivistic oppressors who always take advantage of the treasury and the lesser classes' favor to achieve their purposes and stay in power. When convictions are ideological, they generate crises because they probably, at some point, explained certain circumstances; these forms of expression of thought generate identity, and losing, changing, or expanding them gives rise to a feeling of deep anguish, alarm, or even of danger of a lost identity. This is where the root of Colombian nonconformity is now taking to the streets to demonstrate and delineates an anxiety for change in the political scheme of the nation.

The indescribable, the ineffable, the incomprehensible usually arouse emotion. No reason can be given for emotions except that it is nothing but sensation. The postulates enunciated in the text give account of a structure and of the elements that integrate the process, of structure of mind and thought sign; they allow to direct the gaze towards the emotions from a theoretical perspective, to see its impact on the construction of thought in the interpretant, addressing the sign and its different classifications and relationships. This leads to the emotional intelligence that is enlivened by the opposition sectors through a communication strategy that seeks to fuel the conflict and keep the Protestant mass in the Colombian streets, destabilizing the democratic order of the nation. In a context of social action, subjects intervene with both their knowledge of the world and the emotional motivations that make them incline or participate in such actions. The possibility of interacting is guaranteed by their knowledge of the signs, and the social benefit or performance of that participation is determined by emotions. Without these, there would be no social action.

## 4. Conclusions

The construction of meaning is shaped by the news agenda of the media and the multiple doubts of citizens that are made explicit, on many occasions without having reasons. Individual certainties are not guarantees of truth, so we resort to the formation of communities that consolidate a form of mass action. The digital ecosystem is littered with explicit premises that are scrutinized and involve all possible modes of inference. The reasoning of social problems starts from hypotheses based on the knowledge or introspection of the subject to external things. In a message, a sentence is constituted in the grammatical unit that is part of the formal system of the language, and the statement is the update of the speeches in the communication by the speakers.

The discourse of polarization and ideological confrontation between the left and right materializes the communicative act of Colombians in the media and social sphere. The statement of the message, that which is communicated, cannot be separated from the statement that is the result of the interaction of the sender and the receiver. The act of enunciation leaves traces in the statement, in an environment of conflict, the verbal pronouns "I", "your", "they", among others, take relevance, as do the verbal categories such as person, mode, and time. The content of the messages is endowed with a differential sense on the part of the recipients, its meaning beyond the codes and subcodes of interpretation that are held on the text; that is, the conditions of access to communication and the basic denotation that led to the connotation in the reading journeys that the receiver follows according to his competences, his semantic universes, and his textual repertoires.

The subject's particular experience with communication is divergent, at different times, from that foreseen by the issuer. Communicative interaction brings with it the negotiation of meanings within the various alternatives of codification and decoding that make it possible for texts to be produced and interpreted from different points of view by reference to distinct convention systems. Social platforms as a communication channel determine the use, construction, and organization of the text, materialization of feelings and thoughts

through verbal and nonverbal forms. The network is a meeting place of knowledge in which the interpretation of reality and the possibility of using information depends on the collective representations that govern communicative and social activity.

The worldview of Colombians is conditioned by the ideological beliefs from which the subjects adhere by different affinities: influence of the family group, academic formations, social and professional activities, and critical attitudes, among others. The construction and interpretation of reality is conditioned by various instances of social coexistence: educational institutions, mass media, books; the subjects base their communicative activities on the knowledge they extract from them, and even when they make choices of certain interpretations, they have knowledge of others, whether they provoke distances or social approaches. The proposals of meaning structured by the issuing agent only materialize, or are invalidated, in the interpretation of the messages that it is possible to suppose them and that depend on the interpretive activity of their target poles.

The flow of information in times of crisis is complex because it imposes rhythms and schemes of creation and distribution very distant from the traditional models of action of the media. Although traditional media remain a space for analysis, the participation of communities in the various platforms has gained weight, in which a flow is generated according to the generational and ideological traits of the public of each network or scenario whose relevance already exceeds that of the classic communicative paradigm. In the cases analyzed, we observed the constitution of a biosphere of relationships between the mass media and their users, a mediation process strongly impacted by social networks and affected by the absence of a critical reading of reality by the public. The management of communication in a period of crisis, by issuers and receivers, denotes an offer of contents to the letter of the media industry in convergence with the collective discursive construction of citizens who fight to make themselves heard while claiming to defend democratic values to find solutions that draw solutions to nonconformity. In the communicative ecosystem of Colombians, the construction of journalistic agendas marked by new ways of presenting content is perceived, a communicative framework that requires a cognitive effort on the part of the public to interpret the material circulating on screen.

A post, a tweet, an image, or a video can produce different messages, all of which are possible in the various negotiations that it establishes with social subjects. In the digital platforms, there is a competition of the mass media and opinion leaders to get the attention of users in an ecosystem invaded by a multiplicity of textual manifestations. Markers—still or moving images, hashtags, or keywords—are necessary to achieve effective communication, allow us to know the intentions of the speakers within everyday life, and go beyond what is evident to know how to interpret the voices with which these discourses are generated that have certain degrees of manipulation from the intention with which the speaker transmits.

The digital text is the communicative update of the codes, production is based on conditions such as operational knowledge and communicative intentionality that allow us to update our structures of significance in relation to the situations and contexts of the communicative environment. Codes are not fixed and immutable entities, cultural products are changing and are subject to the constant hybridization demanded by the communicative needs of users. The use subjects its rules to modifications, some momentary, others permanent, and owe their existence to social processes, elements that are manifested in the textual experiences of the subjects, which implies that knowledge and access to them are never homogeneous.

**Author Contributions:** Conceptualization, methodology, validation, formal analysis, investigation, resources, data curation, writing—review and editing, A.B.-R. and G.C.F.V. All authors have read and agreed to the published version of the manuscript.

**Funding:** This research received no external funding.

**Institutional Review Board Statement:** Not applicable.

**Informed Consent Statement:** Not applicable.

**Data Availability Statement:** The data are not publicly available.

**Conflicts of Interest:** The authors declare no conflict of interests.

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
