# Peer review of "The Media in the Construction of Reality in the Context of Colombian Social Nonconformity"

_information, doi:10.3390/info13080378_

Round 1

Reviewer 1 Report

Congratulations to the esteemed authors for their study. I would like to offer a number of suggestions for the development of the study. These are: The method of the study should be clearly emphasized in the abstract section. In the analysis and conclusion part of the study, the findings should be discussed more comprehensively in Colombia. English translations of the Spanish texts in the visuals in the study should be indicated with footnotes. The importance and originality of the current study should be highlighted by making reference to past studies. After these corrections, I find it appropriate to publish the study.

Author Response

In the attached text we respond to the comments of the Reviewer.

Reviewer 2 Report

 The present research is very new as we have not past works related to the field; indeed, the context of lectures is under-explored. The paper can offer insights pertaining to such context. Literature review is adequate. Methodology is clear; precise analyses have been carried out with the research objectives. Findings are fairly presented, and the analysis is presented in a good manner and presenting new ideas. 

Author Response

(The authors gave the same response as above.)
